# Bioinformatic Analysis of Lytic Polysaccharide Monooxygenases Reveals the Pan-Families Occurrence of Intrinsically Disordered C-Terminal Extensions

**DOI:** 10.3390/biom11111632

**Published:** 2021-11-04

**Authors:** Ketty C. Tamburrini, Nicolas Terrapon, Vincent Lombard, Bastien Bissaro, Sonia Longhi, Jean-Guy Berrin

**Affiliations:** 1Architecture et Fonction des Macromolécules Biologiques (AFMB), Centre National de la Recherche Scientifique (CNRS), Aix-Marseille Université (AMU), UMR 7257, 13288 Marseille, France; ketty.TAMBURRINI@univ-amu.fr (K.C.T.); Nicolas.Terrapon@univ-amu.fr (N.T.); vincent.lombard@univ-amu.fr (V.L.); 2Biodiversité et Biotechnologie Fongiques (BBF), French National Institute for Agriculture, Food, and Environment (INRAE), Aix-Marseille Université (AMU), UMR 1163, 13288 Marseille, France; bastien.bissaro@inrae.fr; 3Architecture et Fonction des Macromolécules Biologiques (AFMB), French National Institute for Agriculture, Food, and Environment (INRAE), USC 1408, 13288 Marseille, France

**Keywords:** intrinsically disordered regions, lytic polysaccharide monooxygenase, LPMO, redox-sensitive, conditionally disordered regions, disorder prediction, CAZymes

## Abstract

Lytic polysaccharide monooxygenases (LPMOs) are monocopper enzymes secreted by many organisms and viruses. LPMOs catalyze the oxidative cleavage of different types of polysaccharides and are today divided into eight families (AA9–11, AA13–17) within the Auxiliary Activity enzyme class of the CAZy database. LPMOs minimal architecture encompasses a catalytic domain, to which can be appended a carbohydrate-binding module. Intriguingly, we observed that some LPMO sequences also display a C-terminal extension of varying length not associated with any known function or fold. Here, we analyzed 27,060 sequences from different LPMO families and show that 60% have a C-terminal extension predicted to be intrinsically disordered. Our analysis shows that these disordered C-terminal regions (dCTRs) are widespread in all LPMO families (except AA13) and differ in terms of sequence length and amino-acid composition. Noteworthily, these dCTRs have so far only been observed in LPMOs. LPMO-dCTRs share a common polyampholytic nature and an enrichment in serine and threonine residues, suggesting that they undergo post-translational modifications. Interestingly, dCTRs from AA11 and AA15 are enriched in redox-sensitive, conditionally disordered regions. The widespread occurrence of dCTRs in LPMOs from evolutionarily very divergent organisms, hints at a possible functional role and opens new prospects in the field of LPMOs.

## 1. Introduction

The discovery in 2010 of Lytic Polysaccharide Monooxygenases (LPMOs) revolutionized our vision of the enzymatic conversion of polysaccharides [1]. LPMOs cleave glycosidic bonds by oxidation of the C1 and/or C4 carbon of the scissile glycosidic chain, inducing local disruptions of the polysaccharide network [1,2,3,4,5,6].The fact that in conjunction with glycoside hydrolases (GHs), LPMOs “boost” the hydrolysis of recalcitrant polysaccharides attracted growing attention with many fundamental and applied studies in a biorefinery context [7]. Based on sequence similarity, LPMOs are today distributed in eight AA families (AA9-AA11; AA13-AA17) in the Carbohydrate Active enZymes (CAZy; www.cazy.org) database [8,9]. LPMOs target a wide range of polysaccharides, including chitin (AA10, AA11, AA15), cellulose (AA9, AA10, AA15, AA16), starch (AA13), hemicelluloses (AA9, AA14) and pectin (AA17). LPMOs are found across all domains of life (including bacteria, viruses and also eukaryotes such as yeasts, fungi, oomycetes and animals) and recently functions other than biomass decay have emerged (e.g., microbial pathogenesis) [10,11]. LPMO-like proteins (referred to as X325) have also been identified in fungi and yeasts and are suspected to play a role in copper homeostasis [12,13].

The catalytic domain of all LPMOs hitherto characterized adopts an immunoglobulin-like β-sandwich fold [14]. The active site is exposed at the surface with a mono-copper atom coordinated by a strictly conserved “Histidine-brace”. The catalytic mechanism entails a reduction of the resting Cu(II) state to the active Cu(I) state, followed by reaction of the latter with either O_2_ or H_2_O_2_ to hydroxylate the polysaccharide substrate [15,16]. Like other CAZymes, LPMOs occur either as a single module (i.e., catalytic domain only) or as multimodular proteins. In the latter case, the catalytic domain is connected to either a carbohydrate binding module (CBM) or a glycosylphosphatidylinositol (GPI) anchor (in some AA16 and all LPMO-like X325 members), via glycosylated Ser/Thr/Pro-rich linkers that ensure independent folding and function of the constituent domains [17,18,19].

Due to structural constraints required for catalysis, enzymes adopt a globular and folded conformation. However, there are growing evidences of enzymes (e.g., hydrolases, transferases) possessing functionally-active intrinsically disordered regions (IDRs) [20]. IDRs do not adopt a well-defined conformation and are characterized by a lack of stable secondary and/or tertiary structure in the absence of ligands or partners [21,22,23]. IDRs have increasingly emerged as key players in multiple aspects of biological processes [24]. The failure to adopt a unique, 3D structure is dictated by a peculiar amino acid composition with enrichment in charged, polar and structure-breaking (Pro, Gly) amino acids, and depletion in hydrophobic and aromatic residues, as well as in Cys and Asn [25]. IDRs are able to perform their function either by remaining completely disordered or by undergoing a disorder-to-order transition after interaction with their partner or ligand, a phenomenon known as “induced folding” [26]. Recently, redox conditions have also been shown to play a role in regulating the conformation of so-called conditionally disordered proteins, i.e., proteins whose degree of (dis)order relies on the redox state of their cysteines. In redox-sensitive conditionally disordered proteins, changes in the redox state of cysteines can trigger conformational transitions that ultimately affect protein function [27,28,29]. Although IDRs typically show low sequence identity in alignments, some properties, named ‘molecular features’, may be conserved, such as sequence length [30], amino acid composition [31], charge properties and distribution, sequence complexity and other physico-chemical properties [32]. Conveniently, several computational methods have been developed to accurately predict protein intrinsic disorder and to capture the diverse features of disorder, based on amino-acid sequence [33,34,35].

During our previous investigations, we noticed that some LPMOs display an intriguing C-terminal extension, which was not encountered in other CAZymes. These C-terminal regions are commonly removed during expression construct design prior to LPMO characterization. In this study, we used bioinformatics to analyze these uncharacterized C-terminal extensions occurring in different LPMO families. The objectives were to (i) predict their degree of disorder, (ii) investigate their distribution across LPMO families and organisms and (iii) analyze their molecular features.

## 2. Materials and Methods

### 2.1. Retrieval and Curation of LPMOs Sequences

LPMO amino-acid sequences were retrieved by BlastP [36] searches against the protein sequences of GenBank [37] and of the JGI MycoCosm [38] using biochemically-characterized LPMO sequences as the queries (search performed January 2021, hence not including AA17 family released in August 2021). Sequences from each LPMO family were aligned and manually curated to remove proteins that display a catalytic domain with a truncated sequence, a long insertion or unspecified amino acids. Additional modules and/or motifs (i.e., CBMs, signal peptides, GHs, GPI, etc.) were assigned in a semi-manually annotation process used for the updates of the CAZy database [9]. All those regions that could not assigned to any of these known modules were considered as unknown (UNK). Taxonomic information was also gathered. Redundancy was removed by selecting one representative sequence out of a cluster of >99% identical sequences by using the CD-HIT Suite [39]. With default settings, CD-HIT selects the longest sequence as the representative of each cluster in which shorter sequences are merged. However, since we were interested in analyzing the variation in length of C-terminal extensions, we used the following parameters to obtain representatives of shorter sequences: minimal alignment coverage = 1 for both the longer (aL) and shorter (aS) sequence, and minimal length similarity fraction = 1 (a value of 1 standing for 100%). The number of sequences per LPMO family before and after curation is reported in Appendix A.

### 2.2. Prediction of Disorder of C-Terminal Unknown Sequences

Subsequently to curation, LPMO sequences with a C-terminal region of unknown function longer than 30 residues (UNK) were extracted and analyzed with MobiDB-lite 3.0 [40]. We chose this length threshold as we were interested in analyzing long regions, i.e., regions whose length equals or exceeds the one typically used in large-scale disorder analyses [41,42]. The choice of this predictor reflects the idea of favoring precision over inclusiveness. Indeed, the method ranks among the most precise predictors according to the recent CAID experiment. The latter evaluated 43 disorder predictors, showing that deep learning techniques are the most accurate [33]. Even though MobiDB-lite is highly conservative, it is a standalone method with reduced computing time that allows to capture different flavors of disorder. These features make the predictor appropriate for large scale analyses. Moreover, the predictor is currently used in the InterPro and MobiDB databases to generate disorder predictions [43,44].

MobiDB-lite uses eight different predictors (GlobPlot, three versions of ESpritz, two versions of IUPred, and two versions of DisEMBL) to derive a consensus that is refined to remove short disordered regions and keep only those that consist of at least 20 consecutive residues predicted as disordered (hereafter referred to as Long Disordered Regions, LDRs). Since IUPred and DisEMBL use a 20-residue sliding window, to avoid biases due to the analysis of partial sequences, for each sequence we reconstituted the sliding window by adding the ten amino acids that precede the UNK sequence and used this enlarged sequence for disorder prediction. However, disorder statistics only considered residues strictly belonging to the UNK region, where the boundaries of the latter were inferred as described above. 

A residue is considered as disordered if at least five out of the eight predictors predict it as disordered. A C-terminal sequence was considered to be disordered (and hence classified as dCTR) if MobiDB-lite 3.0 predicts at least one LDR. The fraction of disordered residues is calculated by dividing the number of residues predicted to be disordered by the length of the dCTR. 

### 2.3. Annotation of Features in dCTRs

The properties of the dCTRs were derived from their primary sequence through the toolkit localCIDER [45]. These properties include the length of the sequence, the net charge per residue (NCPR), the fraction of charged residues (FCR), the mixing/segregation of prolines and charged residues compared to all other residues (Ω value) and the diagram-of-states classification. 

The amino acid compositional bias was estimated by the Composition Profiler webserver (http://cprofiler.org/cgi-bin/profiler.cgi) [46], by comparing the content of a given amino acid in the dCTRs of the LPMO family under investigation (C_dCTR_), with the corresponding content in the PDB Select 25 database (C_PDB_), as (C_dCTR_-C_PDB_)/C_PDB_. PDB Select 25 is a data set containing PDB structures with less than 25% sequence identity [47].

### 2.4. Prediction of Redox-Sensitive, Conditionally Disordered Regions

The prediction of redox-sensitive, conditionally disordered regions was generated by the stand-alone version of IUPred2A, keeping default parameters [48]. The method predicts for each sequence two disorder profiles, one for the native sequence and one calculated on the modified sequence with Cys converted to Ser. A redox-sensitive region is considered to occur if the difference between the two profiles is larger than 0.3. The prediction was carried out on the entire sequence of LPMOs bearing a dCTR, but conditional disorder statistics only considered residues strictly belonging to the dCTR. A protein was considered to contain a redox-sensitive region if IUPred2A returned at least one such a region. The minimal length of the predicted region is 15 residues. 

### 2.5. Prediction of Disordered Binding Sites (DiBS)

The prediction of Disordered Binding Sites (DiBS) was generated by the stand-alone version of ANCHOR2 in IUPred3 [49]. ANCHOR2 performed as the best available method in the prediction of DiBS in the CAID experiment [33]. The method uses an energy estimation approach to identify residues belonging to a disordered region and likely to gain stabilization by interacting with a globular partner. The predictor returns a score between 0 and 1 for each residue, representing the probability of that residue to be in a DiBS. The prediction was carried out on the entire sequence of LPMOs bearing a dCTR, but disorder binding statistics only considered residues strictly belonging to the dCTR. A residue was considered to be in a DiBS if the predictor returns a value above 0.5. The minimal length of a predicted DiBS is 5 residues.

### 2.6. Statistical Analysis

Analysis and data processing were conducted by in-house developed Python and R scripts for plotting and performing statistical analyses. The significance of multiple comparisons was evaluated with the non-parametric Kruskal-Wallis test. When the null hypothesis was rejected, we applied the post-hoc Dunn’s multiple comparison test to test for differences among the LPMOs families. The Bonferroni Correction was used to adjust *p*-value for multiple comparisons. 

## 3. Results

### 3.1. LPMOs Contain IDRs, with Family-Dependent Abundance and Modularity

To analyze the C-terminal regions of LPMOs, we parsed 27,060 LPMO sequences, belonging to seven distinct families (AA9-AA11, AA13-AA16). The C-terminal regions of LPMOs (of at least 30 residues in length) were segregated into different categories (Figure 1A): (i) LPMOs without any C-terminal region downstream the catalytic domain, (ii) LPMOs harboring a known CBM or a GH (mostly in AA15 and a few in AA10), (iii) GPI anchors (in AA9, AA14, AA15 and AA16) and (iv) LPMOs with sequences of unknown function that do not share any sequence similarity to known module families (denoted as “UNK”). Some sequences of unknown function were previously defined as X modules based on amino acid sequence conservations [50,51]. We noticed that more than half of AA14, AA11 and AA15 (72%, 56% and 51% of the sequences, respectively) and 45% of AA16 LPMOs display an UNK region. These percentages are lower for AA9 and AA10 sequences (ca. 30% and 13%, respectively), and even more so for AA13 LPMOs (2%) that are however enriched in CBMs (65%). A new LPMO family, AA17 [52], was reported in the literature during the finalization of the present manuscript and was thus not included in our study. In summary, all investigated LPMOs families have, to different extents, UNK C-terminal regions that do not show any sequence conservation.

We then analyzed the LPMO UNK C-terminal regions using MobiDB-lite 3.0 to identify predicted disordered regions. The choice of the MobidDB-lite 3.0, over other disorder predictors, was dictated by the consensus-based approach able to capture different disorder “flavors” and by the implementation of a filtering step that guarantees that the predicted IDR covers at least 20 consecutive amino acids (for reviews on disorder prediction see [33,34,35]), henceforth called LDR. Our analysis revealed that 80% of the UNK sequences in LPMO families AA9, AA14 and AA16, and 60% in AA10, AA11 and AA15 are predicted to possess at least one LDR (Figure 1B). Those UNK sequences were therefore considered as dCTRs and hereafter termed so. 

Considering all LPMOs sequences in the different families (Figure 1C), we noticed that AA14 sequences are particularly enriched in dCTRs (57%), followed by AA16 (36%), AA11 (34%), AA15 (32%) and AA9 (23%). Of note, only 8% of LPMOs of the AA10 family harbor dCTRs. Since LPMOs of the AA10 family are mainly found in prokaryotic species (www.cazy.org), this finding is in agreement with previous studies that have found that prokaryotic proteins are less enriched in IDRs compared to proteins from eukaryotic species [23,53]. Strikingly, the AA13 LPMO family was found to contain only three members with a dCTR, and was therefore not further investigated. 

The modularity of CAZymes is known to play an important role in the function of the enzyme [54]. In light of this, we also looked at the nature of the module preceding the dCTR region (Figure 1D). Among all dCTR-containing LPMOs (henceforth called “LPMO-dCTRs”), single LPMO catalytic modules immediately followed by a dCTR represent the main modularity, and rank as follows: AA14 (100% of sequences), AA16 (100%), AA10 (98%), AA11 (91%), AA9 (88%) and AA15 (82%). A significant number (18%) of AA15 LPMO-dCTRs are multimodular enzymes, harboring glycoside hydrolase (GH9 or GH18) domains, WSC domain (Pfam PF01822) or a combination of different GH and X modules. 8% of AA11 LPMO-dCTRs contain a X278 module between the catalytic domain and the dCTR, where X278 modules were hypothesized to be chitin binding by Hemsworth et al. [47]. We also identified a conserved X280 module in AA9 LPMOs, which is present only in this family and was found in association with a CBM18 chitin-binding domain, suggesting a potential role of X280 in targeting chitin [48]. A few dCTRs of the AA9 family also display a short linear motif (SLiM), annotated as X283 [48]. SLiMs are short regions, occurring within IDRs of eukaryotic proteins, involved in a broad variety of biological functions [55,56]. Of note, dCTRs are rarely found after CBMs (less than 2% in AA10 and AA15, 0.5% in AA11 and 0.6% in AA9).

### 3.2. dCTRs Differ in Length, Fraction of Disordered Residues and Number of Long Disordered Regions

Next, we looked at the peculiarities of the dCTRs (length and amino-acid composition) in the six LPMOs families (AA9–11, AA14–16). While dCTRs can have varying length ranging between 30 (minimum value chosen for this study), to over 1000 amino acids (Figure 2A), the median dCTR length falls within a narrower window, i.e., between 93 (for AA9 members) and 152 residues (for AA11 members). Specifically, 73%, 90%, 60% and 86% of sequences in AA9, AA10, AA14 and AA16 families, respectively, display dCTRs between 50 to 150 residues in length. Instead, dCTRs from AA11 and AA15 LPMOs are longer on average and have a wider variability in length. We also found that 60% of sequences shorter than 50 residues in AA11 are appended to X278 modules. 

Then, we analyzed the distribution of the fraction of disordered residues per dCTR in the families (Figure 2B). Notably, AA9 dCTRs were found to have a significantly higher median fraction of disordered residues (75%) compared to the other families (*p* < 0.01–0.001, Dunn’s test), while AA10, AA14 and AA16 dCTRs have a median disorder content of 70%. AA11 and AA15 dCTRs have a lower disorder content, with a median value of disordered residues of 55% and 45%, respectively. Looking at the fraction of disordered residues as a function of the dCTR length, we observed a general trend, whereby disorder content decreases with the length of the dCTR (Appendix A). In details, we noticed that dCTRs of up to 250 residues have a high disorder content (mean value ≥50%), in all LPMO families except for AA15 LPMOs (Appendix A). In the latter, only chains shorter than 100 residues display a high disorder content (i.e., mean value ≥50%) (Appendix A).

We also analyzed the total number of LDRs in the dCTRs. Appendix A shows that the majority of dCTRs (more than 60% for all LPMO families) have only one LDR. By contrast, dCTRs from the AA15 family often present more than one LDR. This implies that AA15 dCTRs are characterized by alternating disordered and ordered segments.

### 3.3. Most dCTRs Are Enriched in Serine and Threonine Residues

IDRs are characterized by depletion in hydrophobic amino acids and enrichment in polar and charged amino acids [25]. Thus, we compared the composition profile between dCTRs in each LPMO family and the PDB database (Figure 3). We found that dCTRs from all families are depleted in hydrophobic residues and remarkably enriched in Ser, Thr, Pro and Ala residues. Taking into account that Ser and Thr residues are often involved in post-translational modifications (PTMs), and in particular in *O*-glycosylation, the enrichment in these residues in dCTRs could reflect a propensity of these regions to be glycosylated, as already known to happen in linkers connecting different modular CAZymes [19,57,58]. Interestingly, a similar pattern of compositional bias is present in both fungal and bacterial AA10 dCTRs (Appendix A). In spite of the overall similarity in the composition profile of fungal and bacterial dCTRs of the AA10 family, more subtle differences can be discerned: fungal dCTRs are in fact enriched in Cys, Tyr, Met, Arg, Gln, Ser, Asn, Lys, while bacterial dCTRs display more Ala, Gly, Pro, Val, Leu, Asp, Glu, in agreement with the overall composition of low complexity regions in bacteria [59]. Remarkably, dCTRs from the AA11 and AA15 families are particularly enriched in cysteine residues, which could drastically affect the functionality and conformation of these regions [60]. Furthermore, as few as five amino acids (Ser, Thr, Pro, Gly and Ala) constitute on average as much as half of the composition of dCTRs (Appendix A). Yet, there are clear differences in the frequency of these residues between LPMOs targeting glucose-based polymers (AA9, AA14 and AA16), LPMOs targeting chitin (AA11 and AA15), and AA10 (Appendix A). Altogether, these results indicate that dCTRs are characterized by a poor amino acid diversity, a property typical of low complexity sequences, i.e., regions with a biased sequence composition that are frequently found in IDRs. In agreement, low complexity is a widespread and prevalent property in dCTRs across all LPMO families (Appendix A).

### 3.4. dCTRs Are Weak Neutral Polyampholytes with Different Patterns of Charged/Proline Residues

To shed light onto the physicochemical properties of the dCTRs, we analyzed the fraction of charged residues (FCR), the net charge per residue (NCPR) and the position of dCTRs on the diagram of states, as defined by Das et al. [62] (Figure 4A and Appendix A). NCPR corresponds to the difference between the number of positively charged residues and negatively charged residues divided by the total number of residues. FCR defines the fraction of charged amino acids at neutral pH, allowing the differentiation between polyelectrolytes (excess of one type of charge) and polyampholytes (equivalent fraction of opposite charges). High FCR values are associated to a tendency to adopt expanded coil-like conformations, while lower values encode globule-like conformations [63]. The FCR values calculated for the dCTRs are lower than 0.25 in 75% of the sequences in each family, with a mean FCR between 0.13 (AA16) and 0.18 (AA10), and a maximum FCR of 0.4, with few sequences exceeding this value. This finding, along with the NCPR mean value of 0 for all the families (Appendix A), suggests that the dCTRs can be classified as weak neutral polyampholytes. Sequences with FCR and |NCPR| < 0.25 fall in the first region of the diagram of states, adopting globular (i.e., collapsed) conformations. At such low FCR values, patterning of charged residues (κ parameter) is not informative [45], hence we did not carry out the analysis of κ. Figure 4A shows that 90% of dCTRs from AA9, AA10, AA14 and AA16 families, and 80% of AA11 and AA15 dCTRs behave as disordered globules. The remaining sequences are predicted to be chimeras of globules and coils (region 2 in the diagram of states) in 10%, 6.7%, 18.6%, 8%, 14% and 6.7% of AA9-AA11 and AA14-AA16 families, respectively. Between 1% and 4% of the sequences map in the third region of the diagram of states that corresponds to coil conformations, and less than 1% are found in regions 4 and 5. This qualitative conformational prediction is valid if the fraction of proline residues is less than 15% (a condition met by 86% of the dCTRs) [45]. This restriction stems from the fact that a high proline content is associated with more extended conformations than predicted from the FCR and |NCPR| alone. In addition to charged residues, the fraction of proline residues and the patterning of prolines and charged residues may drive local conformational transitions [64,65]. In fact, it is established that enrichment in proline residues expands the conformational ensembles in a way that is not predicted by the FCR alone. To assess residues patterning, we computed the patterning parameter Ω, as defined by Martin et al. [65]. This parameter is normalized between 0 and 1; low Ω values indicate that Pro, Lys, Asp, Arg, Glu are well dispersed across the amino acid sequence, while high Ω values design a segregation of these residues in a specific region of the sequence. Atomistic simulations showed that segregation leads to compaction, while mixing of proline and charged residues engenders expanded conformations [65]. To assess patterning, we selected dCTRs with a content of proline and charged residues equal to or greater than 10%. The Ω distribution across the LPMOs families is shown in Figure 4B. Specifically, dCTRs from AA16 have the highest Ω values (median of 0.4), while AA9, AA10 and AA14 share a similar Ω distribution (median of 0.3). These median Ω values are significantly (*p* < 0.01) different from those observed in AA11 and AA15 families (median values of 0.24 and 0.23, respectively). These Ω values suggest that charged residues are relatively well mixed along the dCTR sequence, thus favoring extended conformation. By contrast, an increase in Ω, as in the dCTRs of the AA16 family, reflects segregation of Pro, Lys, Asp, Arg, and Glu, and favors more compact conformations, as indicated by the representation in the diagram of state (Figure 4A).

### 3.5. dCTRs from AA11 and AA15 Families Display Redox-Sensitive Conditionally Disordered Regions

In light of the redox activity of LPMOs and of the enrichment in Cys in dCTR sequences of the AA11 and AA15 families, we sought to assess whether dCTRs can be redox-regulated, conditionally disordered regions. To explore this aspect, we used a tool implemented in IUPred2A to capture redox-sensitive regions that are likely to undergo disorder-to-order transitions as a function of the redox state of their cysteines [48]. The method is based on the dual behavior of thiols in cysteines that can act as stabilizing/destabilizing agents in IDRs depending on the redox conditions and the presence of disorder promoting residues in the surrounding sequence. Strikingly, the analysis predicts a high fraction of redox-sensitive regions for AA11 and AA15 dCTRs (45 and 51%, respectively, Figure 4C). These regions may respond to changes in redox conditions by undergoing conformational changes, and could thus mediate very rapid and specific post-translational responses. By contrast, redox-sensitive, conditionally disordered regions are predicted to be much less abundant in AA9 (5%), AA10 (1%), AA14 (3%) and AA16 (4.5%) families. The possible biological relevance of this physicochemical peculiarity is further discussed below.

### 3.6. dCTRs Have Disordered Binding Sites

The function of IDRs is often related to binding to other proteins and/or ligands via short recognition elements in disordered binding sites (DiBS). The residues belonging to DiBS have a propensity to form energy favorable interactions with globular partners, instead of intrachain interactions, resulting in at least partial folding of the DiBS. Here, we used the ANCHOR2 predictor to investigate the presence of DiBS in the dCTRs. Figure 5A shows the distribution of DiBSs in the dCTRs of the various LPMO families. For all the families, the majority of dCTRs harbor one DiBS. The highest fraction of dCTRs devoid of DiBSs is observed in members of the AA15 and AA16 family. The large majority of dCTRs—~90% of AA9, AA10, AA11 and AA14 dCTRs, and ~75% of AA15 and AA16 dCTRs—harbor at least one DiBS. While dCTRs from AA9, AA10, AA15 and AA16 members have on average less than 2 DiBS (1.55, 1.41, 1.5 and 1.23, respectively), dCTRs belonging to AA11 and AA14 families have on average 2 DiBS (average value 2.0). This is in agreement with the fact that IDRs tend to have few DIBSs, in contrast with ordered proteins that tend to have multiple binding regions [66]. Moreover, the number of DiBS does not correlate with the length of the dCTR (Appendix A). The fraction of dCTR residues predicted to form DiBS does not depend on the length of the dCTRs, and tends instead to remain constant in AA9, AA11, AA14 and AA15 members (Figure 5B). Interestingly, dCTRs of members of the AA10 family and, to a lower extent, of the AA16 family show a different pattern: the fraction of residues predicted to fall in DiBS increases with the length of dCTRs until 200 residues and decreases beyond this length. Moreover, the median of the length of DiBS in dCTRs of AA10 members is significantly higher (*p* < 0.0001, Dunn test) than that of the other dCTRs (Appendix A). Considering that the median length of dCTRs of members of the AA10 family is 117 residues, these data suggest that AA10 DiBS usually cover all the dCTR sequence. Interestingly, the median of the length of DiBSs harbored by the dCTRs of bacterial LPMOs of the AA10 family is higher compared to that observed in the dCTRs of fungal AA10 LPMOs (Appendix A), a finding in striking contrast with previous observations that pointed out that DiBS are generally shorter in bacteria than in eukaryotes [67].

### 3.7. Distribution of LPMO-dCTRs across Domains of Life, as well as Viruses

As LPMOs are widely distributed in living organisms (and viruses), we investigated the distribution of LPMO-dCTRs across the different kingdoms (Figure 6). LPMO-dCTRs are present in eukaryotes, bacteria and viruses. However, we did not find any LPMO-dCTR in archaea. Noteworthily, the disorder content is somehow similar between eukaryotes and viruses (ca. 0.4–0.7), and strikingly heterogenous within the bacterial kingdom, with a very low disorder content for Bacteroidetes. Among bacteria, a remarkable observation is the presence of LPMO-dCTRs mainly in Actinobacteria (in particular, *Streptomycetales*), characterized by a mycelial morphology. Among the viral LPMO-dCTRs, we noticed the presence of fusolin proteins (AA10 LPMOs), notably studied for their insecticidal activity [68].

### 3.8. Transcripional Regulation and Secretion of Fungal LPMO-dCTRs from Experimental Studies

To get more insight into the role of LPMO-dCTRs, we parsed available experimental data from the literature to know whether genes encoding LPMO-dCTRs are regulated and/or LPMO-dCTRs are secreted. We report in Appendix A and hereafter a few examples [69,70,71,72,73,74,75]. Jagadeeswaran et al. focused on nine AA9 LPMOs genes from *Aspergillus nidulans*, and quantified their transcript levels on different constituents of plant cell wall (i.e., cellulose, xylan, xyloglucan and beta-glucans) [69]. Interestingly, five of these genes encode LPMO-dCTRs with different lengths, some of them being differently regulated on the tested substrates, i.e., AN1041 and AN2388 were significantly induced by pectin while AN3046 was upregulated in the presence of cellulose, biomass sorghum and xyloglucan. Using the coprophilous fungus *Podospora anserina,* a total of seven AA9 out of the 33 AA9 LPMOs encoded by its genome were identified in the secretomes [71]. Interestingly, *Pa*LPMO9D, which was secreted in the sugar beet pulp condition, bears a 90 amino acids-long dCTR. After heterologous expression in *Pichia pastoris*, *Pa*LPMO9D did not show any activity on cellulose [76]. LPMOs from *Gloeophyllum trabeum* are the only example of LPMOs characterized from brown rot fungi, so far. The genome of *G. trabeum* displays four genes encoding five AA9 enzymes, due to alternative splicing of the gene *lpmo*A. In particular, Kojima et al. [72] characterized the variant *Gt*LPMO9A-2, which contains a C-terminal region, herein confirmed to be a dCTR. The enzyme showed a broad specificity and demonstrated a high efficiency in depolymerizing xyloglucans. Interestingly, the authors described four conserved cysteines in the dCTR region, which make this sequence susceptible of redox regulation. In that work, they also described the enzyme *Gt*LPMO9D presenting a low complexity C-terminal region, herein identified as dCTR. The phylogenetic analysis revealed that this enzyme is part of a different clade quite distant from other AA9 characterized so far [72]. Overall, secretomic and transcriptomic data, available from the literature, confirm that genes encoding LPMO-dCTRs are regulated and that LPMOs bearing dCTRs are secreted, although the secretion does not seem to be induced by a specific growth condition.

## 4. Discussion

In this study, using an in-silico approach, we discovered that about 60% of LPMO members, from most of the LPMO families, possess an intrinsically disordered C-terminal region that had hitherto gone unnoticed. Within glycan-processing enzymes, the presence of these dCTRs has so far only been observed in LPMOs. Indeed, to our knowledge, no dCTR has ever been observed in other classes of CAZymes (glycoside hydrolases, carbohydrate esterases, polysaccharides lyases) nor in other types of oxidoreductases (e.g., laccases and peroxidases). Data from the literature report that IDRs can provide several advantages to enzymes, such as ability to tune the affinity for different substrates [77], fast and dynamic regulation of thermodynamic properties, allostery and catalysis [78], stability, subcellular localization [79], and enzyme processivity [80]. Based on the results of our bioinformatic analysis and the current knowledge on LPMOs, we suggest a few roles dCTRs may play. 

The disordered nature of dCTRs mainly arises from an enrichment in Ser, Thr and Pro suggesting PTMs, in particular *O*-glycosylation, which is known to occur in fungal and actinomycetes CAZymes [19,57,58]. The *O*-glycosylation of linkers in CAZymes is known to provide resistance to proteases and to impart extended conformations, thus extending the operating distance of the enzyme thanks to excluded volume effects [81]. It has been suggested that glycosylated linkers can enhance the binding affinity to substrates [19]. Yet, linkers are generally shorter than dCTRs and, by definition, linkers bridge two different modules, while dCTRs in LPMOs are always C-terminally located and usually appended directly after the catalytic LPMO domain. Here, we found that more than 70% of dCTRs are predicted to have at least one disordered binding site, consistent with the involvement of the dCTRs in binding to partners and/or ligands and hence with a function in molecular recognition. These observations, together with the fact that LPMO-dCTRs seem to be differentially regulated and secreted by filamentous fungi in response to plant biomass (Appendix A), could suggest a role of dCTRs in providing promiscuous binding to substrates.

We noticed that bacterial LPMO-dCTRs are mostly found in Actinobacteria from the genus *Streptomyces*. While some *Streptomyces* species efficiently degrade plant biomass, these bacteria also share common morphological traits with fungi, i.e., they form a complex mycelium and sporulate under adverse conditions [82]. Even if direct evidence is missing, it is tempting to speculate that LPMO-dCTRs may mediate attachment to the cell wall, alike the GPI anchor found in AA9, AA14, AA15, AA16 LPMOs and X325 LPMO-like proteins, or the hydrophobic C-terminal LPXTG sequence motif (Pfam PF00746) found in AA10 LPMOs [83]. Interestingly, the role of fungal IDRs in cell-to-cell channels has already been proven to be tied to regulation of the intercellular connectivity [84]. Considering that the fungal cell wall contains chitin, it has been hypothesized that fungal chitin-active LPMOs (e.g., AA11) could be involved in cell wall remodeling and growth. The peculiar feature of AA11 and AA15 dCTRs (redox-sensitive conditionally disordered regions), could be related to this very specific context of cell wall remodeling. Of note, chitin-active AA15 LPMOs are suggested to be involved in chitin remodeling in respiratory and digestive systems of arthropods [85]. Therefore, it would be of interest to consider dCTRs while investigating such hypotheses.

It is also worth considering that LPMO-dCTRs could play a role in pathogenicity although in different contexts. Recent investigations in plant pathogen oomycetes, unveiled a pectin-active LPMO (AA17 family) from *Phytophthora infestans*. The authors observed that the AA17 LPMO catalytic domain is followed by a polypeptide predicted to be intrinsically disordered, which is variable in length and amino-acid composition and rich in Ser, Pro, and Thr residues [52]. This AA17 LPMO-dCTR plays a key role in early stages of *P. infestans* pathogenic cycle, although the contribution of the dCTR in the pathogenic mechanism has not yet been established.

Another intriguing example of a possible connection with pathogenesis, concerns fusolins, which are produced by insect viruses, and are examples of LPMO-dCTRs within the AA10 family. The C-terminal region in this particular AA10 LPMO (with a putative chitin activity) is responsible for spindle formation, i.e., proteinaceous crystalline inclusion bodies which enhance virus infectivity [86]. The crystal structures of different fusolins reveal that the C-terminal regions have various degree of disorder and lack interpretable electron density [68]. Inclusion bodies are sites where viral replication and assembly take place and where specific viral and cellular proteins, as well as nucleic acids, concentrate. Noteworthy, many viral inclusion bodies are liquid-like organelles resulting from liquid-liquid phase separation (LLPS) [87,88,89]. These liquid compartments not surrounded by a membrane can also undergo a “maturation” process towards a more solid-like state, the phenomenon being referred to as “phase transition” [87]. Both processes, which are not unique to viruses and rather constitute a new paradigm in cell biology [90], are driven by proteins bearing IDRs and low complexity regions. Accordingly, the dCTR of fusolins were shown to play a role in driving the formation of insect virus inclusion bodies.

Recently, LLPS has also been shown to play a role in the extracellular milieu, as exemplified by galectin-3 [91]. One of the functions of galectin-3 is to agglutinate glycosylated molecules. This function, in which multivalent galectin-3 acts as a bridge, is lost when its N-terminal IDR is removed. The latter was shown to undergo LLPS and proposed to play a role in the extracellular agglutination of galectin-3 [91]. It is therefore tempting to speculate that the dCTRs might mediate LPMOs LLPS. This process would result in the formation of extracellular biocondensates that could protect LPMOs from oxidative damage. Sequestration inside LLPS droplets, because of the high viscosity, can exclude some interactions (with “external” molecules) and increase some others, by enhancing the probability and kinetics of molecular interactions and enzymatic reactions among “internal” molecules [92]. In this scenario, which awaits experimental validation, the dCTRs would confer a higher enzymatic efficiency to LPMOs. Furthermore, since IDRs are known to undergo PTMs, and since the latter have been shown to regulate LLPS [90], PTMs of LPMO dCTRs (e.g., oxidation/reduction of cysteines and/or glycosylation) could serve as switches in the formation of biocondensates. 

In conclusion, using computational approaches, we have shown that intrinsically disordered C-terminal regions are very common in most LPMO families. The variable length and amino-acid composition of LPMO-dCTRs and their presence across different domains of life (and viruses) may reflect either different functions or a common function conferred by distinct sets of physico-chemical properties that would result from host-specific adaptation. Based on all these in-silico analyses, we call the CAZyme community to include these extensions when characterizing LPMOs in a biological and/or biotechnological context. Further functional investigations are needed to experimentally assess the degree of disorder of these C-terminal regions and provide insights into their role.

## Figures and Tables

**Figure 1 biomolecules-11-01632-f001:**
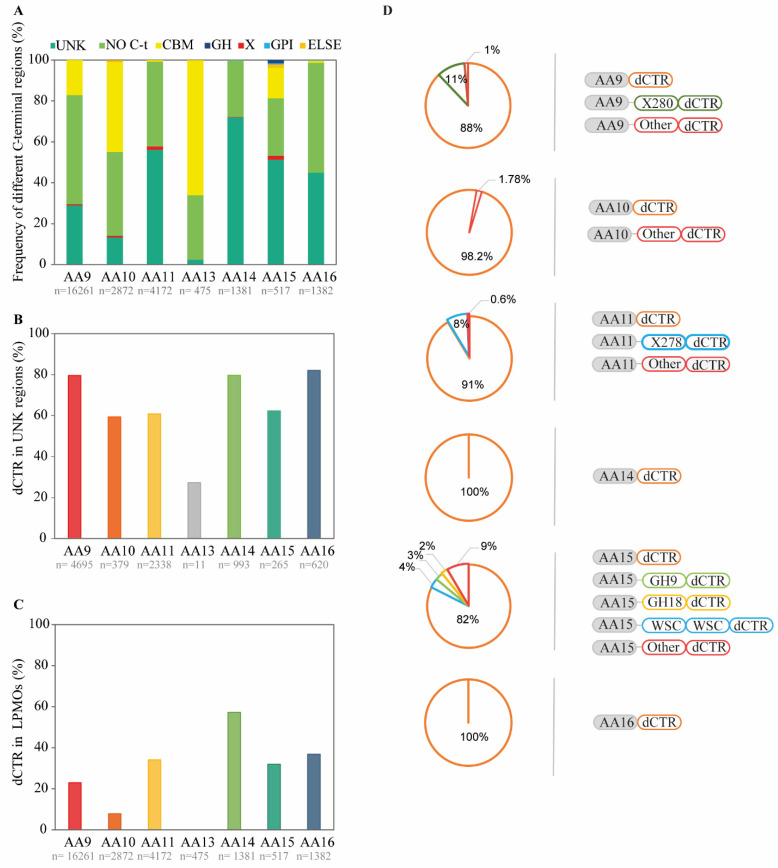
Nature of C-terminal regions occurring in LPMOs. (**A**) Description and abundance of the various C-terminal regions in LPMO families. LPMOs without any C-terminal region downstream the catalytic domain are denoted as “NO C-t”. The “Else” category includes cell wall retention motifs (for AA10, Pfam PF00746) and barwin domains (for AA15; Pfam PF00967). (**B**,**C**) LPMO family-dependent percentage of LPMO-dCTRs among LPMOs with UNK regions (**B**) and among all LPMO sequences (**C**). LPMO-dCTRs are LPMOs predicted to have at least one LDR. “n” is the number of sequences analyzed for each LPMO family. (**D**) Domain architecture of LPMO-dCTRs. Others modules include different CBMs, X domains and combinations of both.

**Figure 2 biomolecules-11-01632-f002:**
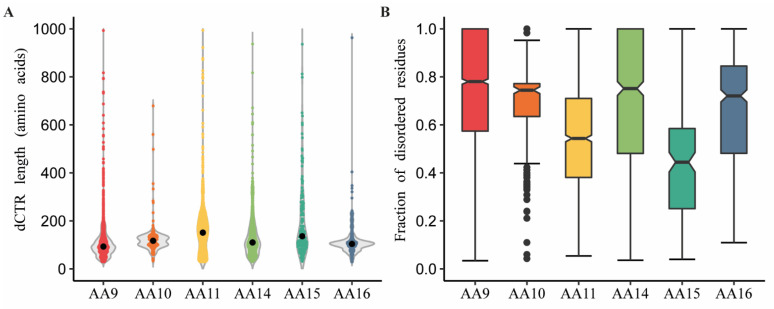
Length and disorder content of the dCTRs. (**A**) Violin plots showing the distribution of the length of dCTRs across the different LPMOs families. The dot in each violin plot represents the median length value. (**B**) Boxplots showing the fraction of disordered residues within dCTRs across the LPMOs families. The central box shows the middle portion (i.e., 25–75%) of the dataset: the bottom line of the box defines the first quartile (25%), the middle line shows the median, and the top line of the box shows the third quartile (75%). The extremities of vertical whiskers represent the largest and smallest values. Outliers are represented by a dot (AA10 family).

**Figure 3 biomolecules-11-01632-f003:**
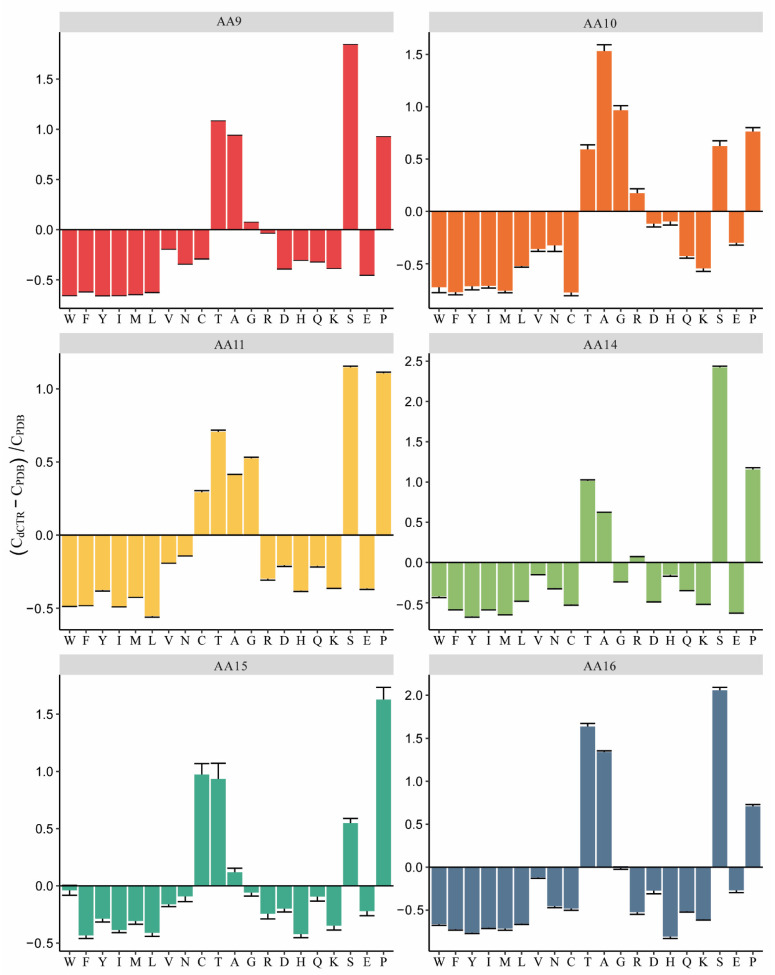
Compositional bias of dCTRs with respect to the PDB database, across the different LPMO families. The fractional difference in the amino acid composition was evaluated as (C_dCTR_-C_PDB_)/C_PDB_, where C_dCTR_ is the content in a given amino acid of all dCTRs of the family under investigation, and C_PDB_ is the corresponding content in the PDB database. The error bars denote the standard deviation over 10,000 bootstrap iterations. Residues have been ordered on the *x*-axis according to the TOP-IDP flexibility index as described in [61]. Plots were generated by the Composition Profiler tool (http://cprofiler.org/cgi-bin/profiler.cgi; [46]).

**Figure 4 biomolecules-11-01632-f004:**
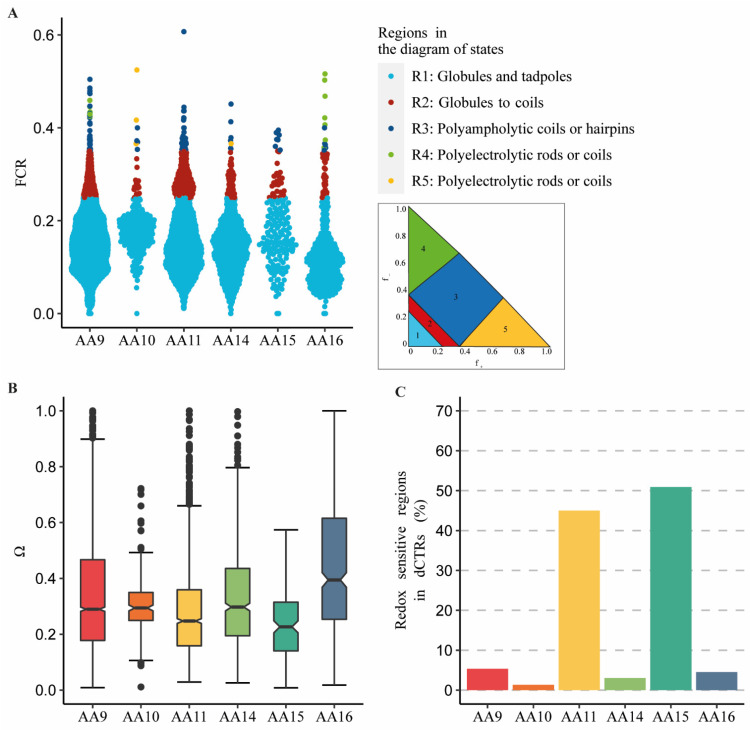
Physico-chemical properties of dCTRs. (**A**) Distribution of fraction of charged residue (FCR) of dCTRs sequences across LPMOs families. Each dot represents a sequence and the color of the dot is associated to the region the dCTR occupies in the diagram of states. The insert shows a representation of the diagram of states. “f_−_“ is the fraction of negative residues, “f_+_” is the fraction of positive residues. (**B**) Boxplots showing the distribution of the patterning parameter Ω values of dCTRs across LPMOs families. The central box defines the first quartile, median, and third quartile from the data and vertical whiskers represent the largest and smallest values on the top and bottom of the boxes. (**C**) Percentage of conditionally redox-sensitive disordered regions predicted by IUPred2A across all dCTRs.

**Figure 5 biomolecules-11-01632-f005:**
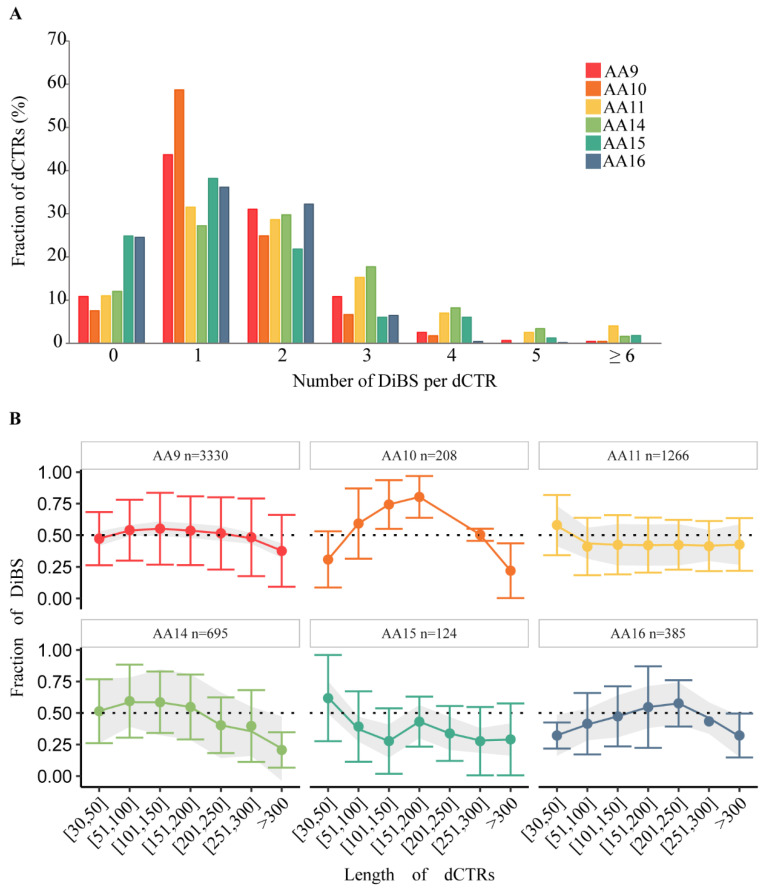
Prediction of Disordered Binding Sites (DiBSs) in dCTRs. (**A**) Number of DiBS predicted byANCHOR2 across the LPMO families. (**B**) Fraction of DiBSs, i.e., number of residues in DiBS divided by the length of dCTRs, plotted as a function of the length of dCTRs across LPMO families, with *n* being the number of sequences per family having at least one DiBS.

**Figure 6 biomolecules-11-01632-f006:**
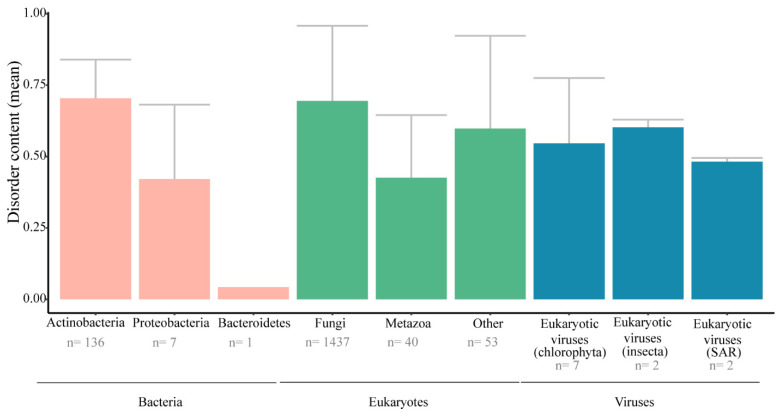
Disorder content of dCTRs in LPMOs across kingdoms. The disorder content is calculated as the mean of the fraction of disordered residues within dCTRs. The eukaryotes category Metazoa includes arthropods and mollusks and the category “Other” includes algae and oomycetes. Abbreviations: SAR, *Stramenopiles, Alveolata, Rhizaria*; *n* is the number of species having LPMO-dCTRs.

## Data Availability

The data present in the current study are available from the corresponding author on reasonable request.

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
