# Peer review of "Bioinformatic Analysis of Lytic Polysaccharide Monooxygenases Reveals the Pan-Families Occurrence of Intrinsically Disordered C-Terminal Extensions"

_biomolecules, 2021, doi:10.3390/biom11111632_

Round 1

Reviewer 1 Report

The authors study several interesting characteristics of the C-terminus of the LPMOs. Their analysis considers propensity for intrinsic disorder, biases in the amino acid composition, net charge, and redox sensitivity. This in-depth analysis is supported by a suite of statistical tests and reveals significant prevalence of intrinsic disorder in these sequence regions. This study is executed very well and, in my view, should be published.

  1. Authors point to [34] as the source to study the topic of disorder prediction. However, this reference is not yet published while numerous already published and recent surveys are available.
  2. While the use of MobiDB-lite meta-predictor can be justified by the ease to collect its results from MobiDB, the authors may want to briefly discuss availability of more accurate methods. The recent community assessment CAID (cited in the article) reveals that the methods applied in this meta-predictor (IUPred, GlobPlot, ESpritz and DisEMBL) provide modest levels of predictive quality, and several more accurate options are available.
  3. Could the low disorder prevalence for Bacteroidetes be caused by the small sample size (only one species), given the large error bars (which I believe represent standard deviations) associated with other types of bacteria in Figure 5 (particularly for Proteobacteria)?

Author Response

The authors study several interesting characteristics of the C-terminus of the LPMOs. Their analysis considers propensity for intrinsic disorder, biases in the amino acid composition, net charge, and redox sensitivity. This in-depth analysis is supported by a suite of statistical tests and reveals significant prevalence of intrinsic disorder in these sequence regions. This study is executed very well and, in my view, should be published.

We thank the reviewer for his/her positive evaluation of our study.

  1. Authors point to [34] as the source to study the topic of disorder prediction. However, this reference is not yet published while numerous already published and recent surveys are available.

Answer: Additional references to reviews on disordered predictions have been added (Refs 33 and 35).

  1. While the use of MobiDB-lite meta-predictor can be justified by the ease to collect its results from MobiDB, the authors may want to briefly discuss availability of more accurate methods. The recent community assessment CAID (cited in the article) reveals that the methods applied in this meta-predictor (IUPred, GlobPlot, ESpritz and DisEMBL) provide modest levels of predictive quality, and several more accurate options are available.

Answer: Following the reviewer’s request we have added a brief discussion of alternative predictors and a more argued rationale for our choice of using Mobi-DB-lite. Specifically, at page 3 one can read “The choice of this predictor reflects the idea of favoring precision over inclusiveness. Indeed, the method ranks among the most precise predictors according to the recent CAID experiment. The latter evaluated 43 disorder predictors, showing that deep learning techniques are the most accurate [33]. Even though MobiDB-lite is highly conservative, it is a standalone method with reduced computing time that allows to capture different flavors of disorder. These features make the predictor appropriate for large scale analyses. Moreover, the predictor is currently used in the InterPro and MobiDB databases to generate disorder predictions [43,44].”

  1. Could the low disorder prevalence for Bacteroidetes be caused by the small sample size (only one species), given the large error bars (which I believe represent standard deviations) associated with other types of bacteria in Figure 5 (particularly for Proteobacteria)?

Answer: The fact that only one Bacteroidetes species out of >20 was found to possess LPMO-dCTRs already suggests that LPMOs in this phylum do not tend to have dCTRs, which is corroborated by the low disorder content observed in the unique case that could be captured by our analysis.

Reviewer 2 Report

This is a very speculative article without any experimental evidence as to the conclusions. The methods applied are absolutely standard. As such, the novelty of the contents of the ms is pretty low. As long as there are no experimental data that support any of the suggestions made in the ms, I do not consider this work ready for publication.

Author Response

This is a very speculative article without any experimental evidence as to the conclusions. The methods applied are absolutely standard. As such, the novelty of the contents of the ms is pretty low. As long as there are no experimental data that support any of the suggestions made in the ms, I do not consider this work ready for publication.

Answer. We agree with the reviewer that standard prediction methods, which already proved to provide accurate predictions, were used in our study. Developing a new methodological approach is beyond the scope of the present study that is meant to raise awareness in the CAZyme/LPMO community of the widespread occurrence of disordered CTRs in LPMOs. The fact that these disordered appendages have been unnoticed hitherto and are not found in other CAZymes participating in plant-cell wall breakdown underscores the novelty and originality of our findings. The awareness of the abundance of disordered C-terminal appendages in LPMOs is expected to foster a new avenue of studies aimed at elucidating their functional role, as opposite to the common trend observed so far consisting in removing such regions prior to LPMO characterization.

Hence while we acknowledge that the present study is a purely in silico study, we think it will pave the way towards future studies aimed at experimentally confirming the predicted disordered nature of the CTRs of LPMOs and at unraveling their function. We already have gathered experimental data showing the disordered nature of one such a CTR both in isolation and in the context of a full-length LPMO (from the AA14 family). These data will be the topic of an article that will be submitted soon. Given the infatuation of the community for the recently discovered LPMO properties in glycan breakdown, our manuscript will focus the attention of the LPMO community and will stimulate experimental projects, carried out not only by ourselves only but also by other groups.

Indeed, LPMO studies have significantly contributed to enlarge our knowledge of CAZymes with many publications/citations in the last decade (e.g. PMID: 29377002, 185 citations (>60 average citations per year)). Of note, pure bioinformatic studies on CAZymes are also highly cited (e.g., PMID: 24270786 cited >4500 times). Finally, we would like to stress that the field of protein intrinsic disorder has also experienced a burst, thanks to a number of seminal, purely in silico studies that have been cited an amazingly number of times (e.g., PMID 22702725, 311 citations (31 average citations per year), PMID 16717195, 315 citations (20 average citations per year), PMID 19596244, 243 citations (19 average citations per year), PMID 12604785, 339 citations (18 average citations per year) to cite a few).

Reviewer 3 Report

This is an interesting and important study dedicated to the analysis of the C-terminal extensions of lytic polysaccharide monooxygenases (LPMOs) that are not associated with any known function or fold. The authors show that these extensions are characterized by the varying length and amino acid composition and are often predicted as intrinsically disordered. Such disordered C-terminal regions (dCTRs) can be found in all LPMO families (except AA13). It is suggested that dCTRs can be of functional importance and can be subjected to various PTMs. The manuscript is well-written and concise. 

In my view, this study should be complemented by the analysis of the potential functional roles of dCTRs using ANCHOR (https://iupred.elte.hu/) or flDPnn (putative function- and linker based Disorder Prediction using deep neural network; http://biomine.cs.vcu.edu/servers/flDPnn/) or some other computational tools for finding disorder-based binding sites.

Author Response

This is an interesting and important study dedicated to the analysis of the C-terminal extensions of lytic polysaccharide monooxygenases (LPMOs) that are not associated with any known function or fold. The authors show that these extensions are characterized by the varying length and amino acid composition and are often predicted as intrinsically disordered. Such disordered C-terminal regions (dCTRs) can be found in all LPMO families (except AA13). It is suggested that dCTRs can be of functional importance and can be subjected to various PTMs. The manuscript is well-written and concise. 

We thank the reviewer for his/her positive evaluation of our study.

In my view, this study should be complemented by the analysis of the potential functional roles of dCTRs using ANCHOR (https://iupred.elte.hu/) or flDPnn (putative function- and linker based Disorder Prediction using deep neural network; http://biomine.cs.vcu.edu/servers/flDPnn/) or some other computational tools for finding disorder-based binding sites.

Answer: Following the reviewer’s comment, in the revised version of our manuscript we have added an analysis of predicted disordered binding sites (DiBSs) using ANCHOR2. We have thus added a new paragraph (2.5) in the Materials and Methods section (page 5), a new paragraph (3.6) in the Results section (page 12), as well as a new figure (Figure 5) and a new supplementary figure (Figure S8). Results showed that more than 70% of dCTRs are predicted to have at least one DiBS, consistent with the involvement of the dCTRs in binding to partner and/or ligands and hence with a function in molecular recognition.

Round 2

Reviewer 2 Report

Dear Authors,

I really agree on your statements about the importance of theoretical studies in many occasions. However, as you state yourself: 'We already have gathered experimental data showing the disordered nature of one such a CTR both in isolation and in the context of a full-length LPMO (from the AA14 family). These data will be the topic of an article that will be submitted soon.'

Why not combine the current ms with your new experimental data?

As of now, I really do not see the potential for the current ms to be published independently.

Reviewer 3 Report

All my critiques were adequately addressed and the manuscript was amended accordingly. I do not have new concerns.